# An Attract-Repel Decomposition of Undirected Networks

## Abstract

Dot product latent space models are a standard method in many areas ranging from social network analysis to computational biology. Such models have issues modeling graphs which include unclosed triangles such as social networks which include latent heterophily (i.e. cases where opposites attract) or co-occurrence graphs which have substitutes (items which occur in similar contexts but not together). We show a minimal expansion to the dot product model which includes both homophily (attract) and heterophily (repel) latent forces. Beyond simply fitting the data, we discuss how to use the AR spaces produced to more deeply understand real networks allowing analysts to measure the latent heterophily in social network formation, detect substitutes in co-occurrence networks, or perform exploratory analysis for candidates for inhibition / activation relationships in systems biology.

Network data is ubiquitous across many disciplines ranging from the natural sciences (Jeong et al., 2001; Barabasi & Oltvai, 2004) to the social sciences (Granovetter, 1985; Easley et al., 2010; Jackson, 2010). However, networks are high dimensional objects and thus can be difficult to work with. Finding easy representations of how nodes are connected is an important topic of study. In the symmetric (or undirected) case a workhorse method are dot product models (Lovász & Vesztergombi, 1999; Ng et al., 2002; Perozzi et al., 2014; Tang et al., 2015; Grover & Leskovec, 2016; Athreya et al., 2017; Lerer et al., 2019). In dot product models each node in a network is associated with an embedding (a.k.a latent vector) in Euclidean space and dot product between vectors reflects the strength of an edge between two nodes.

The dot product model can fail in the sense that while the network has simple structure, the dot product model requires a high dimensionality to represent the network well. Such failure will occur whenever networks exhibit a lack of 'transitivity' $A$ is strongly connected to $B$ and $B$ is strongly connected to $C$ but $A$ is *not* connected to $C$. In such cases the dot product model struggles as representing this triangle requires us to construct vectors where $A$ and $B$ are close, $B$ and $C$ are close but $A$ and $C$ are far. When networks have many such combinations, representing such a network with high fidelity will require high dimensional vectors.

Such 'forbidden triads' (Granovetter, 1973) – or more general versions of this pattern – are ubiquitous in networks. They can occur in social networks where they can be signals of heterophily (individuals with similar attributes are less likely to be friends) or 'enemies' (there is clearly something going on in a citation network if two scientists cite all of the same colleagues but never cite each other). In co-occurrence networks they can indicate that nodes play similar 'roles' such the formation of teams or the combination of ingredients in recipes. Thus, a model which, as an approximation, often closes such triangles (as a low rank dot product decomposition would) is likely obscuring important information in the network connections.

Our key contribution is to study a minimal way to expand the dot product model by considering two sets of latent attributes: ones on which nodes attract and ones on which nodes repel. We refer to this as an attract-repel (AR) decomposition. We show how to construct the 'simplest' $AR$ decompositions from a combination of nuclear norm minimization and eigendecomposition. We then apply the AR decomposition to a series of datasets including social networks, co-occurrence networks, and biological data. We show that the AR decomposition allows simpler (lower dimensional) reconstruction of these networks. In addition, we show that the $R$ space in particular is an interesting object of study by itself.

# 1 Related Work

Recently Seshadhri et al. (2020) show that dot product random models cannot reproduce various macro features of real world social networks. Building on this insight, inspired by work in natural language processing (Mikolov et al., 2013), Chanpuriya et al. (2020) consider factorizing adjacency matrices with two vectors per node, a 'target' and a 'context' vector showing that such embeddings are able to reproduce the macro features. Ruiz et al. (2020) applies the concept of 2 embeddings per item to their dataset of a series of 'baskets' (items purchased at a single time) to find complements and substitutes. One interpretation of our result is that we show the two embedding approach is overparameterized when the underlying matrix is symmetric since the symmetry imposes that for any two vectors $\text{target}_i \cdot \text{context}_j = \text{target}_j \cdot \text{context}_i$. The AR approach takes advantage of this symmetry to give a simpler decomposition.

Non-metric embeddings (i.e. those which cannot be represented as dot product latent spaces) have been studied before Nickel et al. (2011); Van der Maaten & Hinton (2012); Laub & Müller (2004). The above works mostly focus on showing that the non-metric embeddings fit better out of sample, and use local optimization techniques to construct the embeddings. We expand on these works conceptually by showing that these decompositions do not simply 'fit better' but can also be used to use the representation to *understand* the underlying network.

Hoff (2007) studies the 'eigenmodel' which is extremely closely related to the AR decomposition. Our work builds upon this in several ways. First, we formalize many of the heuristic claims made in that paper. Second, we show that the local methods proposed there have multiple solutions and instead gives a method for guaranteed computation of the 'simplest' AR decomposition. Third, and most important we go beyond better out of sample fit as a criterion and show that interpreting the 'negative eigenspace' (in the parlance of the eigenmodel) can lead to many insights about the underlying network structure in networks beyond social networks.

# 2 Dot Product Graph Embeddings

An undirected graph $G$ is a set of nodes $\mathcal{N}$ with cardinality $N$ and edges $\mathcal{E}$. We let $e_{ij}$ be the edge between nodes $i$ and $j$ and allow it to be real valued. Since we will be dealing with symmetric graphs we will talk about $e_{ij}$ though this is also the same as $e_{ji}$. This allows us to encompass a model of a static graph (letting $e_{ij} \in \{0, 1\}$), a generalized random graph model (cite) where $e_{ij}$ is the probability of an edge between $i$ and $j$, or a co-occurrence graph where $e_{ij}$ is a co-occurrence count (e.g. from a co-purchase dataset).

We focus specifically on graphs where the self-edge between a node $i$ and itself is not relevant. This is true, for example, in friendship networks (the concept of whether $i$ is friends with herself is difficult to think about) or co-occurrence networks.

A dot product embedding of a graph is a set of vectors $V$, one for each node $i \in \mathcal{N}$ which we refer to as $v_i$.

**Definition 2.1.** We say that **a dot product embedding represents** $G$ if for any two $i, j$ with $i \neq j$ we have that $v_i \cdot v_j = e_{ij}$.

That is, an embedding represents a graph if the dot product of the vectors gives us the edge weight for all pairs that don't include the self-edge. In applications of these models it is common to use not just the dot product but some link function transformation (Rudolph, 2018), for example in Hoff et al. (2002) the graph is a generalized random graph so $e_{ij}$ is the probability of observing an edge between $i$ and $j$. The probability is modeled via a logistic fucnction so $e_{ij} = \sigma(v_i \cdot v_j)$ where $\sigma$ is the sigmoid function. For the purposes of this paper we stick to the standard dot product to lighten notation though our results can be generalized.

An important question is whether this model is perfectly general. That is, can a dot product embedding represent any graph? The answer is yes:

**Theorem 2.2.** *For any graph $\mathcal{G}$ there exists a family of embeddings $\mathcal{V}$ with generic element $V$ that represent $\mathcal{G}$.*

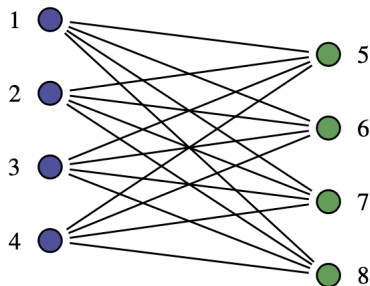

Figure 1: This graph has simple structure but requires high dimensionality to be represented faithfully by a dot product model.

Understanding the intuition behind this result is important for understanding why our proposed AR decomposition is so powerful. Start with a graph $\mathcal{G}$ and construct its adjacency matrix. Note that since the decomposition of $\mathcal{G}$ does specify the self-edges we have that any $M^D$ where $M_{ij}^D = e_{ij}$ for $i \neq j$ and arbitrary diagonal $D$ is a matrix such that if $M^D$ has a factorization $V'V$ these $V$ represent $\mathcal{G}$.

Since we are assuming an undirected graph $G$, any choice of $D$ makes $M^D$ is symmetric. If $M^D$ is symmetric then it has a decomposition of the required form if and only if it is positive semi-definite - i.e. has all positive eigenvalues. The question then becomes: can one always choose a diagonal $D$ for the off-diagonal implied by any $\mathcal{G}$ to make the resulting $M^D$ positive semi-definite?

We can construct one very simply: take the diagonal to be 0. This matrix $M^0$ has all real eigenvalues (since it is symmetric) and either they are all positive (in which case we are done) or not. If there are negative eigenvalues, let $\Lambda$ be the vector of eigenvalues $\lambda$ be the absolute value of the most negative eigenvalue. Now consider the matrix $M^\lambda$ which has the diagonal as all $\lambda$. This matrix can be written as $M^0 + \lambda I$ where $I$ is the identity matrix.

However the means that the $i^{th}$ eigenvalue of the new matrix is $\Lambda_i + \lambda$, in other words. However by construction we know that this vector is weakly positive thus $M^\lambda$ is positive semi-definite. Taking $V$ such that $V'V = M^\lambda$ gives us a dot product embedding of the graph $\mathcal{G}$.

Note that this construction shows us that the embedding is not unique. Rather, for any $D$ which makes $M^D$ positive-semi-definite there is a $V^D$ that can be treated as an embedding of the graph $\mathcal{G}$. By requiring our decomposition to only represent the off-diagonal terms, we are given a lot of freedom.

## 2.1 Issues With the Dot Product Model

While the result above suggests that any graph can be represented by the dot product model, it does not say what dimensionality of the vectors is required. Let $N$ be the number of nodes in the graph. We will define the following property:

**Definition 2.3.** We say that **a dot product embedding $V$ of $G$ is low rank** if it represents $\mathcal{G}$ and the dimensionality of the vectors $V$ is less than $N - 1$.

Note that here we use $N - 1$ as our threshold rather than $N$ since for any graph we know that we can construct the graph Laplacian, which is a positive semi-definite matrix, and thus has a decomposition $V'V$. The graph Laplacian will have a rank of at most $N - 1$ since the dimension of its null space is the number of connected components of the graph, so it is trivial for any graph to construct an $N - 1$ dot product embedding. We say a graph is low rank if we can do *better* than this.

We will now show a graph that intuitively has a very simple structure yet does not have a low rank representation in the dot product model:

Consider the graph in Figure 1. There is a simple structure to the graph: purple nodes connect to green nodes. We say that a dot product model represents the graph if $v_i \cdot v_j = 1$ for all linked nodes and 0 for all unlinked nodes.

Consider the general version of this graph where there are $n_{per}$ nodes of each type. We can show the following result:

**Theorem 2.4.** *There is no low rank dot product embedding of the graph in 1.*

We do not claim to invent this example, a very similar one is used by Hoff (2007) to motivate the eigenmodel. However, our result adds a formal interpretation for what it means for the dot-product model to 'fail' when confronted with this network.

We relegate the full proof to the Appendix, however, we can give the intuition quite simply: the dot product model in low dimensions is 'transitive' in that if $A$ is close to $B$ and $B$ is close to $C$, $A$ must also be close to $C$. In order to express such unclosed triangles the dot product model needs additional free parameters or, in other words, extra dimensions. In the Appendix, we expand this analysis to a popular class of network generating models: stochastic block models (Holland et al., 1983).

## 3   The Attract-Repel Decomposition

The main contribution of this paper is to expand the dot product embedding model to include two latent spaces. Each node lives in two latent spaces: one in which closer implies more likelihood of an edge, one in which closer implies less likelihood of an edge.

We refer to this as an attract-repel (AR) embedding of the graph $\mathcal{G}$. In the case of social networks this translates to familiar forces of homophily (birds of a feather flock together) and heterophily (opposites attract). We denote these vectors $a_i$ and $r_i$ respectively.

**Definition 3.1.** We say that **a AR embedding represents** $\mathcal{G}$ if for any two $i, j$ with $i \neq j$ we have that

$$e_{ij} = a_i \cdot a_j - r_i \cdot r_j.$$

Since a dot product embedding with empty $R$ is an AR embedding (and we know from the prior section that empty $R$ decomposition exists for any graph), the AR problem is over-parametrized: multiple solutions to the problem exist.

In particular, consider the fully heterophilic graph in our counterexample above. It has at least one embedding decomposition where $R$ is empty and $A$ has dimension $2N - 1$. It also has a very simple AR decomposition where $a_i = 1$ and $r_i = 1$ if $i$ is green and $r_i = -1$ if $i$ is purple.

In practice, the existence of multiple solutions with different properties can cause problems when we learn the embeddings from data. Many graph embedding methods use gradient descent or stochastic gradient descent.

Again, here it is worthwhile to look at what our work adds to the eigenmodel in Hoff (2007). That work uses MCMC for optimization. MCMC is a local technique subject to the problem multiple minima and cannot guarantee that given a graph we recover the simplest AR decomposition rather than a 'large' dot-product only decomposition.

### 3.1   Constructing AR Embeddings for a Graph

We will now propose both a way of defining the 'simplest' AR embedding as well a method which is guaranteed to find this solution using provably convergent convex optimization and eigendecomposition rather than a local search.

Letting $\mathcal{AR}$ be the set of all AR embeddings representing a graph $G$ and letting $A$ and $R$ be the stacked embedding vectors, we will define the simplest in terms of Frobenius norm, in other words to solve

$$\min_{(A,R)\in\mathcal{AR}} ||A||_F^2 + ||R||_F^2.$$

Recall our earlier notation of $M_D$ to be the adjacency matrix of the our graph with arbitrary diagonal $D$. The notion of finding the simplest embedding in terms of Frobenius norm of the $A$, $R$ matrices can also be thought of as finding the simplest adjacency matrix diagonal for $M_D$ in terms of the nuclear norm $|| \cdot ||_*$, which is a convex relaxation of the matrix's rank (Candès & Recht, 2009). More formally:

**Theorem 3.2.** *Let $A, R$ be a solution to $\min_{(A,R)\in\mathcal{AR}} ||A||_F^2 + ||R||_F^2$. Let $M_D$ be the solution to $\min_D ||M_D||_*$. Then $M_D = A'A - R'R$.*

We leave the proof to the Appendix as it uses standard techniques from the literature. Importantly, this equivalence tells us a way to construct the AR embedding without using gradient based methods that offer no guarantees of finding the minimum we seek.

1. Construct the augmented matrix $\hat{M}$ whose off-diagonal (OD) is defined by the graph and solve the convex problem: $\min_{\hat{P}} ||\hat{P}||_*$ s.t. $\hat{P}_{ij} = p_{ij} \forall i \neq j$

2. Compute the eigendecomposition of $\hat{P} = Q'DQ$. Any symmetric matrix has such a decomposition with real eigenvalues.

3. (Optional) To use a $k$ dimensional embedding, truncate the $n - k$ smallest (in absolute value) eigenvalues to 0

4. Let $D^-$ be the negative eigenvalues and $D^+$ be the positive ones. Let $Q^-$ correspond to the eigenvectors with negative eigenvalues and $Q^+$ be the eigenvectors with positive eigenvalues.

5. Let $A = Q^+\sqrt{D^+}$ and let $R = Q^-\sqrt{-D^-}$

### 3.2 Proximity in repel space as node 'substitutability'

The interpretation of standard dot product graph embeddings is simple: two nodes with similar vectors have similar neighborhoods. However, with AR embeddings we are able to ask more complex questions.

There is recent interest in using embedding techniques to find substitutable products (Ruiz et al., 2020). Substitutes in this case are defined as products which fulfill the same need - or, in the case of co-purchase graphs, are purchased with the same items but rarer together. For example, both Pepsi and Coke may be purchased with Hamburgers and Fries, but a purchase which contains Pepsi usually does not also contain Coke.

Some popular embedding approaches (e.g. LINE (Tang et al., 2015)) actually produce two separate embeddings one which reflect first-order and one which reflects context similarity and then concatenate them together to use them as node embeddings. A simple way to capture both types of similarity is to train two embeddings per node, a 'target' and a 'context', to learn $p_{ij} = t_i \cdot c_j$. One way to construct such embeddings is via an SVD (or logistic SVD) of the adjacency matrix which constructs both row and column embeddings (Chanpuriya et al., 2020). For symmetric networks this is overparametrized as there are extra constraints: $p_{ij} = p_{ji}$ implies that $t_i c_j = t_j c_i$, which is what gives symmetric decompositions the ability to be written in AR form.

We now show that distance of two nodes in repel space can be interpreted as a measure of their substitutability.

There are two notions of similarity used in graphs. The first is context-similarity, two nodes are **context similar** if they have similar neighbors this is also sometimes referred to as 'role similarity'. We can measure context similarity between $i$ and $j$ by taking

$$C(i,j) = a_i a_j + r_i r_j.$$

The second notion of similarity is the more familiar notion of **first-order** similarity which is how strongly two nodes are connected. We measure this by

$$F(i,j) = a_i a_j - r_i r_j.$$

From these two notions we can define two nodes as substitutes if they have high context similarity but low first order similarity. We can give this a continuous score

$$\text{Substitutability}(i,j) = C(i,j) - F(i,j)$$

which yields $2r_i \cdot r_j$. Since the score is dimensionless, we can simply divide by 2 to get that substitutability is, in fact, similarity in $R$ space.

## 4 Empirical Evaluations

We now consider empirical evaluations of the AR embeddings. In particular our main goal will be to show that the AR decomposition naturally allows us to ask certain questions about properties of the network or of nodes. By contrast, answering such questions via standard dot product embedding approaches may be feasible but is certainly not straightforward.

### 4.1 Measuring Homophily and Heterophily in Real Social Networks

The AR decomposition lets us decompose a social nework into its heterophilic and homophilic parts. In particular, we can compare the variance in $A$ to the variance in $R$ to look at the amount of latent homophily vs. latent heterophily present in the network.

We first consider the anonymized ego-networks (an ego network takes a focal ego, takes all of their friends, and maps the friendships between them) of 627 users of a music social network.[1] We consider users with at least 50 friends (mean ego network size = 81.6).

Denote by $\hat{P}^k$ the $k$ dimensional approximation to $\hat{P}$. We look at the rank $k$ error as $e(k) = \sum_{i \neq j}(\hat{p}_{ij} - \hat{p}_{ij}^k)^2$. The normalized reconstruction error is $\frac{e(k)}{\sum_{i \neq j}(\hat{p}_{ij})^2}$ and the reconstruction fidelity is $1-$ the reconstruction error.

Another choice of diagonal has $d_i = \sum_{j \neq i} A_{ij}$. This is the unsigned graph Laplacian. We look at the off diagonal reconstruction error of this diagonal choice as well referring to it as the Unsigned Spectral Decomposition (USpectral).

We compute the $A$ and $AR$ decompositions using the nuclear norm based procedure described earlier. To compute the approximate nuclear norm minimizing diagonal we use singular value thresholding (SVT Cai et al. (2010)) from the $R$ package *filling* (You, 2020).

We also use the generalized Gabriel bi-cross-validation (BCV) procedure proposed in (Owen et al., 2009) to construct an estimate of each network's optimal approximation rank. In BCV the row and column indices are split into folds, one fold of the matrix, is held out while the rest of the matrix is used to fit a low-rank approximation. The rank chosen is the one which minimizes average held out loss. We use a split of 10 folds. The average BCV chosen normalized rank is approximately 9% of the full rank indicated by a gray line on the plot.

Figure 2 panel a shows the reconstruction error averaged over the 627 networks by the normalized rank ($\frac{k}{\text{Rows}(P)}$) of the approximation. The AR decomposition attains a much lower error than the standard dot product (or A-only) decomposition. The USpectral decomposition does quite poorly at reconstruction of the non-self edges of the graph compared to the AR or even the nuclear norm minimizing A decomposition. However, we note that the Laplacian embeddings have other properties and are not intended simply for optimal representation of the network.

Panel b shows the A vs AR comparison in a different way. We plot how large of an embedding we need to recover a fixed fidelity decomposition on average across the 627 networks. We see that the A decomposition requires a $\sim 50\%$ higher dimensionality to recover the network with the same fidelity as the BCV chosen AR decomposition.

---

[1]These networks were collected via a the network's public API (Rozemberczki et al., 2020) and are available on the Stanford Network Analysis Project https://snap.stanford.edu/data/deezer_ego_nets.html.

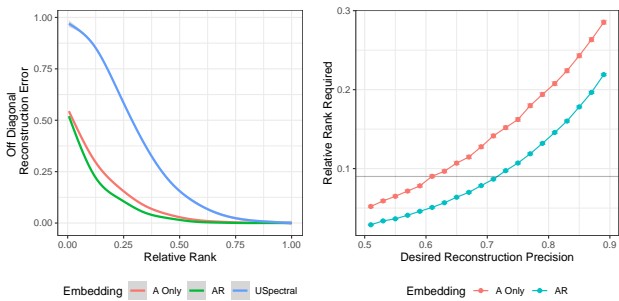

Figure 2: Averaged over 627 real social networks. AR embeddings are much more efficient at reconstructing the network than standard dot product (i.e. *A*-only) embeddings.

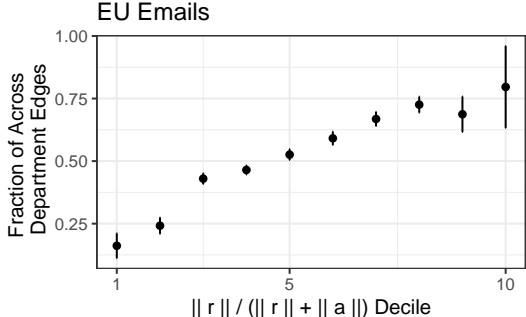

Figure 3: The fraction of an individual's variance explained by negative eigenvalue vectors ($||r_i||$), a.k.a. the individual level latent heterophily, is strongly correlated with a observed measure of heterophily, fraction of e-mail communications outside one's own department.

### 4.2 Measuring Individual Level Heterophily in Social Networks

So far we have focused on the variance explained by the negative eigenvalues as a measure of heterophily in the whole network. This concept can also be applied at the individual level. We consider the EU e-mail network dataset (Leskovec et al., 2007; Yin et al., 2017). This network consists of 1005 individuals at a European research institution. We use a symmetric version of the network where an edge exists between two individuals if they have ever emailed each other.

We construct the *AR* decomposition as above with BCV selecting a 57 dimensional representation. We then compute for each individual the fraction $\frac{||r_i||}{||a_i||+||r_i||}$ where $||\cdot||$ is the $L2$ norm. We argue this can be used to, purely from the graph, measure how likely each individual $i$ is to connect with those 'different' than themselves.

To show that this measure indeed captures heterophily at an individual level, we use the fact that each individual belongs to one of 42 different research departments. Note that the department label is never used in the training of the AR decomposition.

We use fraction of edges outside of own's own department to be one observed measure of heterophily. Figure 3 shows that there is a strong relationship between $\frac{||r_i||}{||a_i||+||r_i||}$ and the fraction of an individual's edges that are outside of their own research department, thus validating our claim that the size of the $r_i$ component captures latent heterophily purely from graph data without the use of any labels.

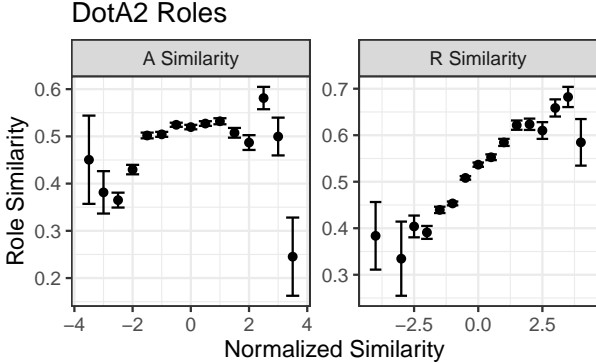

Figure 4: In our DotA data we see that similarity in $A$ vectors does not predict similarity in roles very well but similarity in $R$ vectors (produced without knowing roles) does.

### 4.3 Finding Roles on Teams

We continue to investigate what the $R$ vectors tell us about relationships between nodes. We look at data from the online game DotA2. In this game individuals are placed in a team of 5, each individual chooses one of $\sim 120$ 'heroes', and the team competes against another team of 5.

Heroes in DotA are different and specialized. To be a successful team, a group needs to have a balanced set of heroes. As with many team sports, there are different roles that heroes need to be covered and so real world teams are unlikely to include multiple copies of the same role.

We use a publicly available Kaggle dataset[2] of $\sim 39,000$ DotA matches. From this data we construct a co-occurrence matrix for the heroes. Letting $c_{ij}$ be the co-occurrence between $i$ and $j$. Because the co-occurences are extremely right skewed, we consider the matrix of $\log(c_{ij} + 1)$ though qualitatively all our results go through using the raw co-occurrence counts as well.

We take the $AR$ decomposition of this co-occurrence matrix. We again use the BCV procedure to select the dimensionality of the embedding. The BCV procedure selects a 10 dimensional representation and we find that 5 of these dimensions are associated with negative eigenvalues and 5 are associated with positive ones.

We now show that the $R$ component in the decomposition of the co-occurrence matrix is precisely capturing the notion of role similarity. In other words, heroes that are close by in $R$-space occupy similar roles. As with the example above, we use real role labels which are not used during any training to evaluate the claim.

In DotA2, the general thinking is that each hero can fill one or more of 9 possible roles. These roles are Carry, Disabler, Durable, Escape, Initiator, Jungler, Nuker, Pusher, and Support. See `https://dota2.fandom.com/wiki/Role` for more details. Heroes usually can fill more than one role, though role types are correlated (for example, most Nuker heroes are usually not Durable).

We take what are (as of the time of this paper) the roles the DotA-wiki states that each hero can play and construct a 9 dimensional vector with a 1 if the hero can play that role and a 0 otherwise. We then compute the cosine similarity between the role vectors of any two heroes. Let $\theta_{ij}$ denote this cosine similarity. A higher cosine similarity means that heroes occupy more similar roles.

Figure 4 shows that there is a strong correlation between the $\theta_{ij}$ and $r_i \cdot r_j$ but not so much between $\theta_{ij}$ and $a_i \cdot a_j$, showing that the $R$ component can be used to detect role similarity purely from co-occurrence data.

---

[2]Available here: `https://www.kaggle.com/c/mlcourse-dota2-win-prediction/overview`.

| target | substitute | score |
|--------|-----------|-------|
| baking mix | bisquick | 0.78 |
| baking powder | baking soda | 0.85 |
| beer | apple juice | 0.40 |
| brown sugar | sugar | 0.74 |
| buttermilk | skim milk | 0.52 |
| chicken broth | vegetable broth | 0.63 |
| lemon | fresh lemon juice | 0.76 |
| onion | scallion | 0.71 |
| orange juice | honey | 0.61 |
| parmesan cheese | mozzarella | 0.64 |
| parsley | dried parsley | 0.59 |
| pecan | walnut | 0.85 |
| pecan | sliced almond | 0.65 |
| red wine | dry white wine | 0.68 |
| unsalted butter | margarine | 0.67 |
| unswtd chocolate | baking cocoa | 0.74 |
| vegetable oil | canola oil | 0.88 |
| vinegar | cider vinegar | 0.89 |
| yogurt | greek yogurt | 0.70 |

Table 1: Substitutes for various focal ingredients found by looking at cosine similarity neighbors in the $R$ component.

### 4.4 Substitutes in Ingredients

We now investigate a different task. We use a dataset of $180,000+$ recipes available on Kaggle[3]. We construct the log co-occurrence matrix of the 1000 most common ingredients in these recipes. We compute the AR decomposition of this matrix using the same methodology as the experiment above. We use the rank chosen by BCV ($k = 125$).

We then look at some commonly substituted cooking ingredients.[4] We did not use all examples on the site for two reasons. First, some of them were not in the top 1000 most commonly used ingredients. Second, some substitutions explicitly require a mixture of multiple (3 or more) other ingredients, which is not achievable in the current version of our model.

In Table 2 we take some focal ingredients and show their nearest $R$ neighbors using the cosine similarity. We include only the top neighbor for space here, in the Appendix we include an expanded version of the table including the top 3 suggested substitutes per target ingredient. We see that using $R$-similarity as a substitutability metric seems to yield qualitatively good results in this dataset.

We see that some ingredients have good substitutes while others do not. For example, nearest neighbors of many items have $R$ similarities between .7 and .9 and seem sensible (canola oil for vegetable oil or greek yogurt for yogurt or baking cocoa for unsweetened chocolate). However, beer's closest $R$ neighbor is the less sensible apple juice with much lower similarity scores (around $.3 - .4$).

Nevertheless, we see that $R$ similarity, which is high when context similarity is high but first order similarity is low, seems to be a good way to detect substitutable nodes in co-occurrence networks.

### 4.5 Inhibition and Activation in Biological Networks

Systems biology is a field focusing on study of interactions between genes or proteins. Deterministic or stochastic dynamical systems are usually used to model these interactions. However, the topology of the governing equations is quite often partially or fully unknown.

Existing measurement techniques such as mass cytometry Bendall et al. (2012) or single cell RNA sequencing Luecken & Theis (2019) produce a snapshot of which proteins or genes are active/present in a given cell. Unfortunately, it is difficult (and often mathematically impossible) to recover the full governing structural equations just from such snapshots.

However, just from the snapshot we may be able to glean partial information. In particular, we can consider what it means for gene $A$ and $B$ to be active in the same contexts but never together: such patterns can occur when $A$ and $B$ lie on the same (or closely related) pathways but inhibit each other. Thus, even though real data comes from a directed graph, recovered $R$ similarities may help locate particular kinds of inhibitor pairs.

---

[3]The dataset is available at `https://www.kaggle.com/shuyangli94/food-com-recipes-and-user-interactions`, it originally appeared in Majumder et al. (2019)

[4]The list from which we draw is available at `https://www.allrecipes.com/article/common-ingredient-substitutions/`.

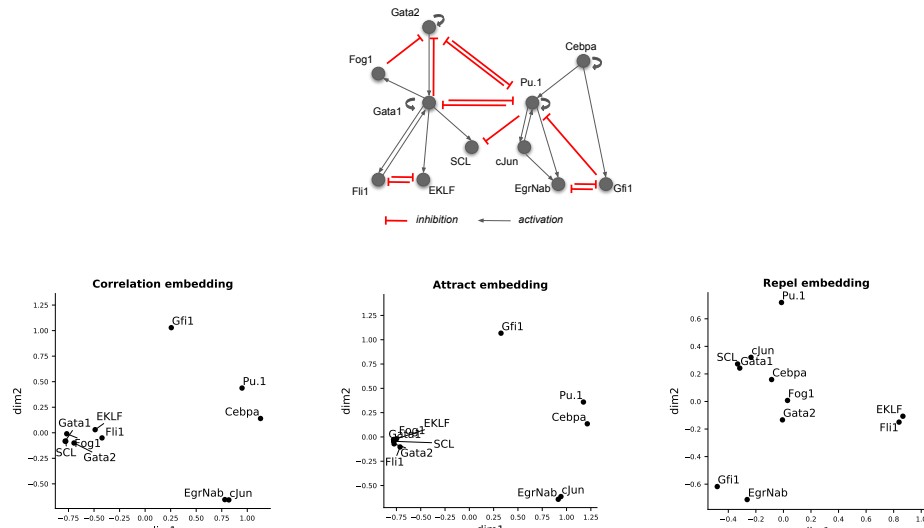

Figure 5: Top panel shows the true directed network generating the symmetric co-occurrence patterns we observe with red indicating inhibition and black denoting activation. Other panels show a kernel PCA of the correlation matrix dot product mode as well as the $A$ and $R$ components respectively. While the inhibition/activation structure is well preserved in the $AR$ embeddings, it is not nearly as clear in the standard 'attract only' decomposition.

Though it is worth noting that this is an important loss of information: inhibitor is a directed relationship but $R$ similarity (or indeed any notion gleaned from a correlation matrix) is a symmetric relation.

Most gene regulatory networks are not completely understood, so commonly simulations of a gene regulatory networks are used. In our final example we use a simplified model of hematopoietic stem cell differentiation Krumsiek et al. (2011).

This network consists of 11 transcription factors and 28 regulatory interactions, some of which exhibit mutual activations and some of which exhibit inhibition (see 5). The original network is represented by boolean rules, which we translated into a system of ODEs. We then sampled observed snapshots of expressions from the system. We construct the co-occurrence matrix of transcription factors using Spearman correlation. The co-occurrence matrix (with original diagonal) is PSD so it can be factorized as $V'V$.

We take this 'correlation embedding' as the baseline dot product embedding. We also compute the AR decomposition of the correlation matrix using the same methodology as the experiments above.

In Figure 5 we use Kernel PCA to visualize the standard dot product embeddings (panel B), $A$ similarities (panel C) and $R$ similarities between the transcription factors (panel D).

We see that mutual inhibitors have the highest $R$-similarity scores (and so are close together in the Kernel PCA representation). One way inhibitions have lower scores, which is not surprising, because the inhibitions doesn't happen immediately and at some points of time both transcription factors can still be observed together. A similar story is obvious in the $A$-similarities showing mutual and one-way activations. However, looking at the embedding of the correlations, such relationships are not obvious. This is partially driven by the fact that in the dot product embedding it is hard to differentiate between two items which have similar contexts but do not appear together (i.e. inhibitors) from items which have low correlation because they are on very different pathways.

We can show the impression above another way in Figure 6 - we can compute normalized similarities either from the pure correlation embeddings (a purely dot product decomposition) as well as $R$ and $A$ similarities for different types of pairs: activators, inhibitors, and unconnected pairs. As above, we see that $R$ similarities

appear to be useful for distinguishing inhibitor pairs which, at least in this small network, often have the property that they appear in similar contexts but not together.

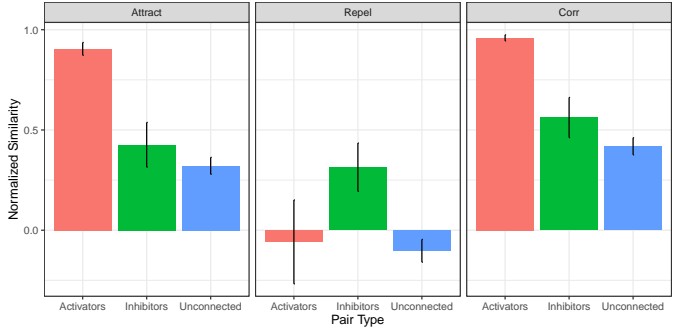

Figure 6: Cosine similarities either from the pure correlation embeddings (a purely dot product decomposition) as well as $R$ and $A$ similarities for different types of pairs: activators, inhibitors, and unconnected pairs.

## 5   Conclusion

Dot product latent space models are a standard method in many areas ranging from social network analysis to computational biology. We have shown that these models are not very good at modeling graphs which include heterophily (i.e. cases where 'similar' types are unlikely to link/appear together). We have shown a minimal expansion which includes both homophily (likes attract) and a heterophily (likes repel) forces. We have shown that this model allows us to ask questions that the standard dot product models cannot in a variety of domains ranging from social networks to systems biology.

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

## 6 Appendix

### 6.1 Proof of Theorem 2.4

In essence, we want to determine the rank, that is, determine the dimension of the nullspace of a matrix

$$M = \begin{pmatrix} A & R \\ R & B \end{pmatrix}$$

where $A$ is an $n \times n$ diagonal matrix with coefficients $a_1 \ldots a_n > 0$, $B$ is an $n \times n$ diagonal matrix with coefficients $b_2 \ldots b_n > 0$, and $R$ is an $n \times n$ matrix of all 1. Let's solve!

$$M \times \begin{pmatrix} \mathbf{u} \\ \mathbf{v} \end{pmatrix} = 0 \quad \Longleftrightarrow \quad \forall i \begin{cases} a_i u_i + \sum_j v_j = 0 \\ b_i v_i + \sum_j u_j = 0 \end{cases}$$

Since $a_i > 0$ and $b_i > 0$, this implies

$$\forall i \quad u_i = -\frac{1}{a_i} \sum_j v_j \quad v_i = -\frac{1}{b_i} \sum_j u_j \tag{1}$$

If we were free to set $\sum_j v_j$ and $\sum_j u_j$ as we please, the two equations equation 1 would describe a 2-dimensional space. Therefore the nullspace of $M$ has dimension at most 2. However we can also use the first of these equations to write

$$\sum_i u_i = -\sum_i \frac{1}{a_i} \sum_j v_j \tag{2}$$

Therefore our nullspace has dimension at most 1. But we can continue and use the second equations from equation 1 to replace $v_j$ above:

$$\sum_i u_i = -\left( \sum_i \frac{1}{a_i} \right) \left( \sum_j v_] \right) = \left( \sum_i \frac{1}{a_i} \right) \left( \sum_j \frac{1}{b_j} \right) \sum_k u_k$$

Therefore, if $r = \left( \sum_i \frac{1}{a_i} \right) \left( \sum_j \frac{1}{b_j} \right) \neq 1$, then we must have $\sum_i u_i = \sum_j v_j = 0$ which means that $u_i = v_j = 0$: the matrix is nonsingular. On the other hand, if $r = 1$, then I can choose $\sum_j v_j$ equal to any non zero value, deduce $\sum_j u_i$ using equation 2, compute $u_i$ and $v_i$ using equation 1, and verify that we have described a one-dimensional nullspace.

In conclusion: if $r = \left( \sum_i \frac{1}{a_i} \right) \left( \sum_j \frac{1}{b_j} \right) \neq 1$, the matrix has full rank. If $r = 1$ the matrix has rank $2n - 1$. This is the case, for instance, when $a_i = b_j = n$.

### 6.2 Relationship to Stochastic Block Models

We now look at the relationship of the $AR$ decomposition to a popular model in the literature: the stochastic block model (SBM) (Holland et al., 1983). These models are used in network analysis in fields as different as looking at human social networks to modeling protein interactions (Airoldi et al., 2006; Abbe, 2017). The SBM works as follows: there are $d$ blocks. Each node belongs to one block.[5] For blocks $i$, $j$ there is a probability $b_{ij}$ that a node from $i$ is connected to a node from $j$. Let $B$ be the matrix of node connection probabilities. We assume $B$ has rank $d$. Let $P_B$ be the implied matrix of probability of connections in a population of $n$ nodes. Again, ignore the diagonal.

It is known that if $B$ is positive semi-definite and rank $k$ then there exists an $A$-only decomposition of rank $k$ which represents the SBM.

We can generalize this result to the $AR$ decomposition as well as generalize our example in Figure 1:

---

[5]The extension to a mixed membership model (Airoldi et al., 2008) is straightforward and would not change our main result.

**Theorem 6.1.** *If $B$ has rank $d$ then $P_B$ has the following properties:*

1. *$P_B$ has AR decomposition of rank $d$*

2. *If $B$ is positive semi-definite then a dot product embedding of $P_B$ can have rank $d$*

3. *If $B$ has negative eigenvalues then a dot product embedding of $P_B$ has rank $> d$.*

*Proof of Theorem 6.1.* We first prove that $P_B$ has an $AR$ decomposition of rank $k$. To do this, we will explicitly construct the decomposition. Since $B$ is symmetric and real it has an AR decomposition of $B = XX' - YY'$. We give each node the embedding corresponding to the embedding of it's block. This is an $AR$ decomposition of $P_B$.

From this the second point follows directly.

The third point follows by contradiction. Suppose that $P_B$ has a dot product embedding of rank $d$ (or less). Let this be $V$. By construction it must be that all nodes with the same block $k$ have the same embedding, call it $v_k$. But then $B = V'V$. But this implies that $B$ is positive semi-definite, which we know it is not. □

**Lemma 6.2.** *The following SBMs will not be positive semi-definite:*

1. **Any SBM with heterophily across blocks:** *There exist some blocks $i, j$ where nodes in $i$ are more likely to connect to $j$ than to $i$ and nodes in $j$ are more likely to connect to $i$ than $j$.*

2. **Any SBM with a triangle:** *There are $3$ blocks $i, j, k$ with the properties that*

$$b_{ii}(b_{jj}b_{kk} - b_{jk}^2) - b_{ij}(b_{ij}b_{kk} - b_{jk}b_{ki}) + b_{ik}(b_{ij}b_{jk} - b_{jj}b_{ik}) < 0.$$

*Proof of Lemma 6.2.* By Sylvester's Criterion for a full rank matrix to be positive semi-definite, any principal minor (submatrix of the same indexed rows and columns) must have non-negative determinant. A $2 \times 2$ submatrix of $i, j$ has the determinant

$$b_{ii}b_{jj} - b_{ij}^2$$

where the square is there due to symmetry of $b_{ij} = b_{ji}$. If $b_{ij} > b_{ii}, b_{jj}$ this determinant is negative meaning that $B$ cannot be positive semi-definite and thus must have negative eigenvalues.

The proof for the second statement in the Lemma is the same, since the equation in the Lemma is in the fact that determinant of a $3x3$ principal minor. □

The three-wise case is particularly interesting as we will see later that unclosed triangles play a special role in co-occurrence networks (note the condition in the Lemma is satisfied with a basic triangle where $i, j$ connect to $k$ but not to each other $b_{ii} = b_{jj} = b_{jk} = 1$ and $b_{ij} = 1, b_{jk} = 1$ but $b_{ij} = 0$).

It is possible to construct higher order interaction conditions that guarantee that $B$ is not easily $A$ decomposable but these conditions become harder to interpret.

## 6.3 Proof of Theorem 3.2

By Lemma 6 of Mazumder et al. (2010) we know that for a matrix $X$ we can write

$$||X||_* = \min_{UV=X} \frac{1}{2}(||U||_F^2 + ||V||_F^2||).$$

With the minimum being attained at the factor decomposition $X = UV$.

By construction of our matrix $\tilde{M}$ it has the factor decomposition $U = [A, R]$ and $V = [A, -R]$ where $[\cdot]$ denotes column-wise concatenation.

| target | substitute | R score |
|---|---|---|
| baking mix | bisquick | 0.78 |
| baking mix | biscuit mix | 0.73 |
| baking mix | bisquick mix | 0.70 |
| baking powder | baking soda | 0.85 |
| baking powder | whole wheat flour | 0.51 |
| baking powder | all-purpose flour | 0.42 |
| beer | apple juice | 0.40 |
| beer | mango | 0.39 |
| beer | corn oil | 0.39 |
| brown sugar | sugar | 0.74 |
| brown sugar | honey | 0.69 |
| brown sugar | light brown sugar | 0.68 |
| buttermilk | skim milk | 0.52 |
| buttermilk | soymilk | 0.48 |
| buttermilk | chickpea | 0.39 |
| chicken broth | chicken stock | 0.85 |
| chicken broth | vegetable broth | 0.63 |
| chicken broth | vegetable stock | 0.61 |
| lemon | fresh lemon juice | 0.76 |
| lemon | lemon, juice of | 0.71 |
| lemon | lemon juice | 0.66 |
| onion | red onion | 0.71 |
| onion | scallion | 0.71 |
| onion | yellow onion | 0.68 |
| orange juice | honey | 0.61 |
| orange juice | orange | 0.50 |
| orange juice | lemon | 0.47 |
| parmesan cheese | mozzarella | 0.64 |
| parmesan cheese | cheddar | 0.62 |
| parmesan cheese | olive oil | 0.53 |
| parsley | fresh parsley | 0.93 |
| parsley | flat leaf parsley | 0.65 |
| parsley | dried parsley | 0.59 |
| pecan | walnut | 0.85 |
| pecan | nut | 0.75 |
| pecan | sliced almond | 0.65 |
| red wine | dry red wine | 0.79 |
| red wine | dry white wine | 0.68 |
| red wine | white wine | 0.61 |
| unsalted butter | butter | 0.74 |
| unsalted butter | margarine | 0.67 |
| unsalted butter | heavy cream | 0.48 |
| unswtd chocolate | unswtd choc square | 0.83 |
| unswtd chocolate | baking cocoa | 0.74 |
| unswtd chocolate | unswtd cocoa | 0.71 |
| vegetable oil | oil | 0.96 |
| vegetable oil | canola oil | 0.88 |
| vegetable oil | olive oil | 0.67 |
| vinegar | cider vinegar | 0.89 |
| vinegar | white vinegar | 0.87 |
| vinegar | apple cider vinegar | 0.82 |
| yogurt | plain yogurt | 0.73 |
| yogurt | greek yogurt | 0.70 |
| yogurt | vanilla yogurt | 0.53 |

Table 2: Substitutes for various focal ingredients found by looking at cosine similarity neighbors in the $R$ component.

Substituting the definition of the factors gets

$$||\tilde{M}||_* = \frac{1}{2}(2\sum_{ij} a_{ij}^2 + 2\sum_{ij} r_{ij}^2)$$

which simplifies to

$$||\tilde{M}||_* = ||A||_F^2 + ||R||_F^2.$$

Recall the construction of $\tilde{M}$ is by finding the smallest $A, R$ in terms of $||A||_F^2 + ||R||_F^2$ to fit all the off diagonal elements. Thus, these $A, R$ also solve the constrained nuclear norm minimization problem.

## 6.4 Expanded Ingredient Substitution List

In the main text we reported the top $R$ neighbor for each focal ingredient to save space. Here we report the top 3 neighbors per each focal ingredient.

