# OpenReview forum: "An Attract-Repel Decomposition of Undirected Networks"
_TMLR — Rejected by TMLR_

### Review · Reviewer_copN · 2022-09-16

**Summary Of Contributions:**

This paper proposes a attract-repel model to extend the dot product model to heterophilic and homophilic networks. It proves the feasibility through theoretical analysis and gives the procedures for constructing the attract-repel model. Extensive experiments on real-world networks show the effectiveness of the proposed method.

**Broader Impact Concerns:**

I have no concerns on the ethical implications of the work.

**Requested Changes:**

The paper overall is clear and well-structured. The following suggestions would strengthen the work.
1. In the experiment part, it would be better to introduce more about the methods used and why we use them. E.g., SVT, BCV, Uspectral.
2. Add some comparison with other heterophily baselines.


**Strengths And Weaknesses:**

Strengths
1. It proposes an attract-repel model that makes the dot product latent space model available on heterophilic graphs and co-occurrence graphs.
2. It gives a way to find the analytic expression of the attract-repel model without gradient descent.
3. It gives two notions of similarity used in graphs, context similarity and first-order similarity, to measure the attract-repel.
4. An abundant empirical analysis of the properties of various types of real-world networks exists.

Weaknesses
1. Except for metrics like reconstruction error, do we have some results on graph tasks, like node classification and link prediction? How can I get the final representation through A and R matrices if I want to do the node classification task?
2. The paper states that it works on heterophilic graphs. How is the performance compared with other baselines that address the heterophily problem? (https://arxiv.org/pdf/2202.07082.pdf, https://proceedings.neurips.cc/paper/2020/file/58ae23d878a47004366189884c2f8440-Paper.pdf )
3. The experiment part is a little unclear about how you got the properties and why you use these methods. E.g. SVT, BCV, Uspectral.

---

### Review · Reviewer_NQkg · 2022-09-19

**Summary Of Contributions:**

In this paper, the authors propose a node embedding algorithm which can capture both homophily and heterophily in a network. The proposed approach first solves a convex optimization problem where the nuclear norm of a matrix is minimized. It then performs the eigenvalue decomposition of the emerging matrix and produces two types of embeddings using the positive and negative eigenvalues and their associated eigenvectors. The authors evaluate the proposed method in several tasks such as in measuring homophily and heterophily in social networks, in finding roles in an online game, in detecting inhibitor pairs in biological networks and others.

**Broader Impact Concerns:**

There are no major ethical concerns.

**Requested Changes:**

- As discussed above, it has to be made clear how exactly the information present in the A and R embeddings is different from the information present in embeddings that capture first and second order node proximity, respectively. In addition, why one should use the proposed embeddings instead of the ones that emerge from algorithms that capture first and second order node proximity?

- The authors should give some examples of node embedding algorithms that are well established in the field of graph representation learning and that are instances of the dot product model. Otherwise, the authors should identify the limitations of embedding algorithms which perform a decomposition of the form $V^\top V$ (many well-known algorithms fall into this category).

- I would suggest the authors experiment with some standard datasets and compare the proposed approach against some baselines. For instance, there are some standard homophilic networks on which the proposed model could be evaluated such as cora and citeseer. There are also some heterophilic datasets that are publicly available such as Texas, Wisconsin and others. Moreover, to validate the proposed approach, it is of paramount importance to compare its performance against that of some recent baseline methods.

- In p.5, step 1 of the algorithm, there is the following constraint: $P_{ij} = p_{ij} \forall i \neq j$. What does $p_{ij}$ represent? I would expect the constraint to be $P_{ij} = e_{ij} \forall i \neq j$ where $e_{ij}$ is the weight of the edge between the $i$-the and $j$-th node.

- In p.5, $M_D$ should be replaced with $M^D$ just to be consistent with previously introduced notation.

- In p.3, the authors mention that the graph Laplacian has a decomposition $V^\top \, V$. While this is true, the values of the non-diagonal elements of the Laplacian matrix are not equal to those of the adjacency matrix (the non-zero elements). Thus, decomposing the Laplacian matrix does not actually fall into the family of dot product models as defined by the authors. Thus, I would suggest the authors provide another example of method to support the argument about the dimension.

- In p.5, the authors mention that "there are two notions of similarity used in graphs". This is not true. There are many more definitions of similarity. I would suggest the authors rephrase this sentence. In addition, they mention that context similarity is equivalent to role similarity. I also do not think this is true. Role similarity usually depends on the structure of the neighborhoods of the nodes. For instance, two nodes can be structurally similar even if they are far apart in the network, but the structures of their neighborhoods are similar to each other (similar ego subgraphs).

- The abstract should be followed by an introduction section (just add a header to the first part of the paper).

- Even though the paper is easy to read, there are a few typos:\
p.2: "two embedding approach is" -> "two embedding approaches are"\
p.2: "is friends" -> "is friend"\
p.11: "and a heterophily" -> "and heterophily"

**Strengths And Weaknesses:**

Strengths
--
- In my view, the idea behind the paper is interesting. The proposed method is simple but possibly new in the graph representation learning literature. There are other node embedding algorithms that learn two different types of latent representations (e.g., deepwalk, node2vec), however, to the best of my knowledge, the formulation in this paper is different from prior work.

- I really liked the experimental section. The paper features a very extensive experimental evaluation. The authors have mainly focused on showing that the proposed approach can actually capture heterophily and substitutability. The experimental results seem promising, however, as mentioned below the proposed method is not compared against any baselines.

Weaknesses
--
- Throughout the paper, the authors claim that the proposed embedding approach can capture both homophily and heterophily. However, it is not clear to me how exactly the information present in the A and R embeddings is different from the information present in embeddings that capture first and second order node proximity, respectively. I would suggest the authors give more details about that in the paper.

- Section 2 presents some limitations of the dot product model, e.g., there are graphs for which no low rank dot product embedding exists. Those results hold for approaches that factorize the adjacency matrix (and which ignore its diagonal values) as follows: $V^\top V$. However, recent node embedding approaches are not instances of this model. These approaches (such as deepwalk and node2vec) use more than one embedding matrices, e.g., they employ a factorization of the form $U^\top  V$. Therefore, I feel that section 2 describes the limitations of some approaches that either do not exist or are not used in practical settings.

- Even though as mentioned above, many different experiments are presented in section 4, several of those experiments are qualitative. More importantly, the proposed model is not compared against any baseline method. Since there are no baselines, it is hard to tell whether the proposed approach is useful and whether it can capture any properties of networks that existing approaches cannot.

---

> ### Author Response · Authors · 2022-09-27
> **Reply (Part 1)**
>
> We thank the reviewer for their careful reading. We discuss what we understand to be two broad points first and discuss the more specific comments afterward.
>
> Please note that we have not updated the main text of paper as the TMLR guidelines ask authors to wait until all 3 reviews have been submitted (see https://www.jmlr.org/tmlr/editorial-policies.html).
>
> --Point 1: VV’ and VU’ embeddings--
>
> The reviewer makes a good framing of the problem: recent literature talks about the difference between VV’ embeddings and VU’ embeddings. So, what is the relationship of AR to these papers?
>
> First, note that methods which give a VU’ factorization of an undirected graph can be rewritten as an AR decomposition by simply taking the approximated matrix (which is symmetric by assumption) induced by VU’ and doing an AR decomposition of it. This happens because, as we point out, the VU’ decomposition is, in some sense, overparameterized since symmetry implies v_i u_j = v_j u_i.
>
> In these cases the relevant matrix to AR-approximate may not be the adjacency matrix, for example, logistic SVD says $p(e_{ij} = 1) = \sigma (v_i u_j)$ so the matrix to decompose is the matrix of logits VU’ is the one that has the linear AR form.
>
> A similar logic applies to the case of the GNN since we can always take the implied proximity matrix between nodes and it should be symmetric (if the underlying graph is undirected), so we can take the AR decomposition of this graph to measure implied heterophily. In other words, any GNN is “linearizable” using AR.
>
> This “refactorization” seems like the most natural way to answer questions about substitutability or homophily/heterophily from any symmetric graph embedding, especially from an unintepretable black box GNN.
>
> We are happy to be clearer on these point, as well as delineate more clearly which existing approaches are of the VV’, versus VU’ variety.
>
> --Point 2: More datasets/baselines--
>
> The goal of the paper is not to try to “beat” existing algorithms in downstream node classification or held out link prediction. We are not wedded to optimizing the square loss on the adjacency matrix (as in the discussion above of how AR relates to VU embeddings or GNNs), rather we choose it because it is the simplest way to a) illuminate the conceptual points of the paper, and b) shows how the AR decomposition captures heterophily at the graph or node level or properties of node pairs (e.g. substitutability) in the R component *without using labels or node features* during training.
>
> Indeed, we are not sure of even what a “baseline” here since the only way we can think of to interrogate a GNN about what it thinks are substitutable nodes (or how much heterophily there is) is using the AR approximation on the GNN pairwise-node similarity matrix as described above.
>
> However, we agree with the referee that more experiments is always better, especially with well known datasets.
>
> Given that the main point of the AR work is to learn about heterophily/substitutes in an simple manner, we can incorporate extra datasets as follows:
>
> Recall that in the current version of the paper in experiment 1 we use $||A|| / (||A|| + ||R||)$ as a measure of latent homophily in a network and in experiment 2 we use this at an individual level to show that $||a_i|| / (||a_i|| + ||r_i||)$ is a good proxy for how homophilous an individual node is.
>
> We can expand this analysis to Cora, Citeseer, EU, Wisconsin, Texas, Cornell, wiki-Squirrel, and wiki-Chameleon. In recent work on heterophilous GNNs (e.g. Luan et al. https://arxiv.org/pdf/2109.05641.pdf) Cora/Citeseer are used as homophilous baselines and Wisconsin, Texas, Cornell, wiki-Squirrel, wiki-Chameleon are said to be heterophilous datasets.
>
> We perform two analyses.
>
> First, we look at whether the $||A|| / (||A|| + ||R||)$ at the network level, aka. the latent variance explained in each network by it’s A component, predicts network level heterophily. We see that it lines up with Luan et. al.’s characterization via GNNs as well as proxying homophily by proportion of edges between labeled nodes (note that some nodes have missing labels in some of the networks).
>
> We also perform the analysis at an individual node level. We median split the individuals into “high” and “low” R by $||r_i|| / ||a_i|| + ||r_i||$. That is, high R individuals are those whose vector is "mostly" R.
>
> We see that that individuals with high R portions have more links to differently labeled nodes than those with more A-like vectors across all datasets (we do not include Squirrel or Chameleon since those datasets have a continuous label) . Thus, we have an individual level proxy for heterogeneity in homophily learned purely from network structure.
>
> When all reviews are in and we updated the paper we will add these analyses in graphic figures/include standard errors, for now we copy/paste the summary tables for each of these in the next comment.

---

> > ### Author Response · Authors · 2022-09-27
> > **Reply (part 2)**
> >
> > Replies continued...
> >
> > -- Other Comments --
> > Re: typos: We will fix the issues with subscripts and typos.
> >
> > Re: Laplacian. We use the *unsigned* Laplacian, this is the adjacency matrix on the off-diagonal and the degree on the diagonal. This has a V’V decomposition which is what we refer to as the “unsigned spectral” embedding. Note that we also include the nuclear norm minimizing V’V decomposition as another example of a V’V embedding, which is the natural “ablation” for the AR.
> >
> > Re: “many notions of similarity in graphs”. We did not mean that these are the *only* two notions of similarity ever discussed, we can change the text. We note that the term “structural equivalence” is used in different ways across the sociological literature, not always with a mathematical definition accompanying it. We are happy to just remove this discussion as it is not central to anything in the paper and simply talk about how second and first order similarity capture different notions of closenesss (and how the AR decomposition relates to this).
> >
> > ----
> > Table results of experiments described above.
> >
> > Graph level analysis:
> >     network / Latent Homophily / Label Homophily / Luan et. al. Score
> >
> > 1  squirrel /0.57 / Labels not discrete /  0.02
> >
> > 2 chameleon / 0.59 / Labels not discrete / 0.06
> >
> > 3 wisconsin/ 0.60 / 0.16 / 0.09
> >
> > 4     texas / 0.66/ 0.06/ 0.01
> >
> > 5   cornell/ 0.69 / 0.11 / 0.05
> >
> > 6  citeseer / 0.81 / 0.72 / 0.62
> >
> > 7      cora / 0.82/ 0.83/ 0.76
> >
> > 8        eu/ 0.88/ 0.46 / Not featured in Luan et al.
> >
> > Individual level analysis:
> >     network / Low R Fraction / High R Fraction
> >
> > 1  citeseer      1.641509       2.158491
> >
> > 2      cora      1.113065       2.434343
> >
> > 3   cornell      5.125000      13.090909
> >
> > 4        eu     24.011327      36.929412
> >
> > 5     texas      6.500000      17.100000
> >
> > 6 wisconsin      4.928571       9.593750
> >
> > Outcome = # of edges to nodes with other labels

---

### Review · Reviewer_kBz3 · 2022-10-31

**Summary Of Contributions:**

The authors focus on dot product latent space models, which learn representations that mimic node similarity (e.g. edges). They propose an Attract-Repel (AR) extension which enables to jointly learn two node representations: one where nodes attract and ones where they repel. This is achieved via a combination of nuclear norm minimization and eigendecomposition. The authors show that their method enables better dealing with specific cases where existing dot product methods struggle, such as detecting substitutes in co-occurrence networks or measuring heterophily in social networks.

**Broader Impact Concerns:**

No concerns have been identified.

**Requested Changes:**

Overall I believe that the paper needs further development.  All the requested changes have been mentioned in the 'Weaknesses' part above. To summarize, changes are critical in the way that the different aspects are presented, and in the experimental evaluation. Besides, I have also expressed concerns about the novel aspects of the proposed methodology.

**Strengths And Weaknesses:**

**Strengths.**

* The paper tackles an actual limitation of dot product latent representation methods, and the intuition provided in the paper is interesting.
* It proposes a simple but sound method to improve upon this specific class of methods. It has also given some concrete examples of where the proposed approach could benefit, or at least where such an approach may fail.

**Weaknesses.**

***About the context and problem tackled.***
* Details about the context of the paper are not provided. The paper presents a representation learning method without giving any context on the field. It focuses on dot product latent representation methods but does not refer to Matrix Factorization or Random Walk based approaches once in the paper, which I believe is essential given they are exactly the type of approach for which this functioning is relevant. Going further, the paper does not situate the work with respect to current GNN methods, which are now commonly used to perform all sorts of graph classification tasks. It is therefore very difficult to situate the proposed work, especially when little background or intuition is provided about dot product-based methods. More generally, the related work is insufficient. The dedicated section is too small and without a clear structure. I would suggest to clarify the structure and to expand drastically the number of approaches investigated, to provide a proper context. Finally, I would encourage the authors also to consider seminal papers in the field ([1], [2]) and some relevant survey articles ([3], [4]).

* The paper makes relevant remarks on heterophily and other limitations of the dot product method. But this problem has already been tackled in the literature. Some approaches also explore constructing node representations preserving structural roles (for instance, [5], [6], [7]). Such models should be discussed in the paper, also indicating how their approach is different from the one presented in the paper.

* The definition of some concepts is not very clear. The structure of the first sections is a bit confusing. Also, similarity measures between nodes, approximated by the dot product of node representations, can take different forms and is not always the existence probability of an edge, which is specific.

***Methodology***

* On the technical side, the contributions and novelty of the proposed methodology are limited. The method is also described too quickly. The paper should further elaborate on negative eigenvalues, the negative product $r_i r_j$, and the pipeline actually utilised in experiments. For instance, introducing BVC here instead of section 4.1, while it is used for all experiments to determine the rank (and is rather essential).

***Experiments.***

* This section is the one that needs further development in my opinion. The paper describes various use cases and performs some small experiments on them, but it does not make some concrete conclusion in the end. The paper should state clearly what it aims to prove first, and then conduct thorough experiments to demonstrate it. Furthermore, the paper does not compare the proposed methodology to any baseline, there is no mention of scalability or running times, and many details about the experiments seem to be omitted. All these points should be taken into consideration to further improve the empirical analysis of the proposed methodology.

* Here are some further comments that could help to enhance this part further.
  - In Section 4.3: please compare your work with other role similarity embedding methods.
  - In Section 4.2: please use metrics for heterophily, that already exist in the related literature (e.g., [8], [9])
  - In Section 4.4: please provide a clear result and not a description of some targeted instances, from which we it's not easy to draw any conclusion.

***Minor comments.***
* Notation: it would be helpful for the reader to follow a convention on the notation. For instance, vectors in lower-case bold, matrices in upper-case bold, scalars in normal font.
* Refine the description of some examples (e.g., the one of the co-occurrence network).
* Make the argument about citation networks and the intuition of dot product approaches stronger.
* The term "undirected graph" is more commonly used than “symmetric” graph, which I did not know. You may want to make it explicit.
* In Figure 1, although we understand why, it should explicitly be mentioned why dot product methods struggle.

***Suggested bibliography.***

* [1] Grover, Aditya, and Jure Leskovec. “node2vec: Scalable feature learning for networks.” Proceedings of the 22nd ACM SIGKDD international conference on Knowledge discovery and data mining. 2016.
* [2] Perozzi, Bryan, Rami Al-Rfou, and Steven Skiena. “Deepwalk: Online learning of social representations.” Proceedings of the 20th ACM SIGKDD international conference on Knowledge discovery and data mining. 2014.
* [3] Hamilton, William L., Rex Ying, and Jure Leskovec. “Representation learning on graphs: Methods and applications.” arXiv preprint arXiv:1709.05584 (2017).
* [4] Zhang, Daokun, et al. “Network representation learning: A survey.” IEEE transactions on Big Data 6.1 (2018): 3-28.
* [5] Wang, Xiao, et al. “Community preserving network embedding.” Thirty-first AAAI conference on artificial intelligence. 2017.
* [6] C. Donnat, M. Zitnik, D. Hallac, and J. Leskovec, “Spectral graph wavelets for structural role similarity in networks,” arXiv:1710.10321, 2017.
* [7] L. F. Ribeiro, P. H. Saverese, and D. R. Figueiredo, “struc2vec: Learning node representations from structural identity,” in Proc. 23rd ACM SIGKDD Int. Conf. Knowl. Discovery Data Mining, 2017, pp. 385–394.
* [8] Li, You, et al. “Graph Representation Learning Beyond Node and Homophily.” IEEE Transactions on Knowledge and Data Engineering (2022).
* [9] Li, Shouheng, Dongwoo Kim, and Qing Wang. “Restructuring Graph for Higher Homophily via Learnable Spectral Clustering.” arXiv preprint arXiv:2206.02386. (2022).

---

### Decision · Action_Editors · 2023-01-06

**Recommendation:** Reject

**Comment:**

All three reviewers (kBz3, NQkg, copN) ultimately leaned towards rejection.

On the positive side,
- Addressing an actual limitation of dot product based methods was appreciated [kBz3].
- The idea/intuition behind the paper was considered interesting [kBz3, NQkg], and making a dot product latent model available for heterophilic and co-occurrence graphs was appreciated [copN].
- The proposed method was considered simple [kBz3,NQkg] but sound [kBz3] and possibly new [NQkg], and the analytical expression of the attract-repel model was appreciated [copN].
- Some reviewers appreciated the experimental part, considering it abundant [copN] and extensive and results promising [NQkg].

However, on the negative side,
- It was criticized that the dot product limitations may not be present in recent node embedding approaches [NQkg], and the lack of addressing previous approaches tackling limitations of dot product methods was criticized [kBz3]
- Lack of results on graph tasks, like node classification and link prediction was criticized [copN] and the lack of experimental comparison against baselines was criticized [kBz3,NQkg,copN].

There were several additional criticims. Reviewer kBz3
- Criticizes lack of sufficiently broad positioning of the work with respect to representation learning approaches such as matrix factorization, random walk, and graph neural network approaches.
- Notes lack of clarity in places.
- Considers contributions and novelty limited.
- Criticizes lack of concrete overall conclusions from the experiments, lack of scalability details, and lack of various experimental details.

Similarly, reviewer NQkg
- Desired improved treatment of prior embedding algorithms.
- Criticizes lack of clarity in how the information present in the different embeddings differs.
- It was criticized that graph Laplacian decomposition would not fall into the defined dot product model family.
- The claim of there being only two notions of similarity was criticized.

Although authors attempted to respond to two of the reviewers [NQkg,copN], ultimately several issues remained. Thus at the current state the work does not seem ready for TMLR.

**Audience:**

(see below)

**Claims And Evidence:**

(see below)

---

> ### Author Response · Authors · 2023-01-06
> **A bit confused**
>
> We are a bit confused that the rejection came before we were able to upload a reply to the last review and the revised paper. As there was no deadline, we chose to take time to make sure all of our experiments were bulletproof rather than upload something hasty. In addition, as all 3 reviewers commented on GNNs we spent quite a bit of time including a section dealing with GNNs and link prediction.
>
> Note that the last review came at the beginning of November, asked for major revisions, and the holiday season occurred.
>
> We also point out that the TMLR policies ask us to wait before we upload any revised paper before all 3 reviews are in as well, so while we already had ready several extra experiments in September responding to reviewers NQkg, copN (see our replies) we could not upload a revised version of the paper.